# Identification of a Novel Variant in Myelin Regulatory Growth Factor by Next-Generation Sequencing Led to the Detection of a Clinically Inapparent Congenital Heart Defect in a Patient with a 46,XY Disorder of Sex Development

**DOI:** 10.3390/jpm13071158

**Published:** 2023-07-19

**Authors:** Lourdes Correa Brito, Romina P. Grinspon, Jimena Lopez Dacal, Paula Scaglia, María Esnaola Azcoiti, Agustín Izquierdo, María Gabriela Ropelato, Rodolfo A. Rey

**Affiliations:** 1Centro de Investigaciones Endocrinológicas “Dr. César Bergadá” (CEDIE), CONICET–FEI–División de Endocrinología, Hospital de Niños Ricardo Gutiérrez, Gallo 1330, Buenos Aires C1425EFD, Argentina; lcorrea@cedie.org.ar (L.C.B.); rgrinspon@cedie.org.ar (R.P.G.); jlopezdacal@cedie.org.ar (J.L.D.); pscaglia@cedie.org.ar (P.S.); mesnaola@cedie.org.ar (M.E.A.); aizquierdo@cedie.org.ar (A.I.); gropelato@cedie.org.ar (M.G.R.); 2Unidad de Medicina Traslacional, Hospital de Niños Ricardo Gutiérrez, Buenos Aires C1425EFD, Argentina; 3Departamento de Histología, Biología Celular, Embriología y Genética Facultad de Medicina, Universidad de Buenos Aires, Buenos Aires C1121ABG, Argentina

**Keywords:** 46,XY DSDs, whole exome sequencing, *MYRF* gene, meso/dextrocardia, OCUGS

## Abstract

In patients with 46,XY disorders of sex development (DSDs), next-generation sequencing (NGS) has high diagnostic efficiency. One contribution to this diagnostic approach is the possibility of applying reverse phenotyping when a variant in a gene associated with multiple organ hits is found. Our aim is to report a case of a patient with 46,XY DSDs in whom the identification of a novel variant in *MYRF* led to the detection of a clinically inapparent congenital heart defect. A full-term newborn presented with ambiguous genitalia, as follows: a 2 cm phallus, penoscrotal hypospadias, partially fused labioscrotal folds, an anogenital distance of 1.2 cm, and non-palpable gonads. The karyotype was 46,XY, serum testosterone and AMH were low, whereas LH and FSH were high, leading to the diagnosis of dysgenetic DSD. Whole exome sequencing identified a novel, heterozygous, nonsense variant in *MYRF*, classified as pathogenic according to the ACMG criteria. *MYRF* encodes a membrane-bound transcriptional factor expressed in several tissues associated with OCUGS syndrome (ophthalmic, cardiac, and urogenital anomalies). In the patient, oriented clinical assessment ruled out ophthalmic defects, but ultrasonography confirmed meso/dextrocardia. We report a novel *MYRF* variant in a patient with 46,XY DSDs, allowing us to identify a clinically inapparent congenital heart defect by reverse phenotyping.

## 1. Introduction

The finding of ambiguous external genitalia in a newborn prompts immediate medical attention. Under usual conditions, testes differentiate from the gonadal ridges in the 46,XY embryo and produce testosterone and anti-Müllerian hormone (AMH), which are responsible for the virilisation of the internal and external genitalia. In the 46,XX embryo, ovaries differentiate and do not secrete androgens or AMH; thus, the internal and external genitalia undergo the female pathway [1]. When the appearance of the genitalia is not typically male or female, the diagnostic process of a disorder of sex development (DSD) is initiated. According to the karyotype, DSDs can be classified into 46,XY DSDs, 46,XX DSDs, or chromosomal DSDs [2]. In 46,XY individuals, insufficient virilisation reflects impaired androgen biosynthesis or action. Gonadal dysgenesis is characterised by low androgen and AMH production, isolated androgen synthesis defects present with low androgens and male-range AMH, and impaired androgen action or androgen insensitivity occurs with male-range androgen and AMH levels [1]. Chromosomal DSDs usually result in gonadal dysgenesis. In 46,XX DSDs, there is excessive androgen production responsible for the virilisation of the fetus, with congenital adrenal hyperplasia as the most frequent aetiology [2]. 

Except for the rare cases due to exposure to exogenous or maternal hormones, most DSDs are of genetic aetiology [3,4]. When the affected gene is expressed exclusively in the gonads, the resulting phenotype is limited to the reproductive organs. Conversely, when it is more ubiquitously expressed, associated malformations usually result in other organs, which may drive the diagnostic process. Amongst the various strategies for genetic diagnosis, next-generation sequencing (NGS) technologies have the highest diagnostic efficiency [5]. Furthermore, these techniques based on massive parallel sequencing introduce the possibility of identifying genes causing DSDs with associated malformations in nonreproductive organs. In these cases, reverse phenotyping represents an advantage in the search for inapparent or neglected anomalies.

Our aim was to report the case of a newborn with dysgenetic 46,XY DSDs in whom the detection of a novel gene variant in *MYRF* (Myelin Regulatory Factor; OMIM 608329) allowed the identification of a clinically inapparent congenital heart defect.

## 2. Materials and Methods

### 2.1. Clinical Assessment and Diagnostic Testing

Length was measured using an infantometer, and weight was determined with a calibrated scale and expressed as a standard deviation score (SDS) based on the Argentine population reference [6]. Physical examination performed by a paediatric endocrinologist included the assessment of external genitalia according to the External Genitalia Score (EGS) [7]. Serum follicle-stimulating hormone (FSH), luteinising hormone (LH), testosterone, 17-hydroxyprogesterone, androstenedione, oestradiol, and AMH were measured using validated assays, as previously published [8,9,10]. Peripheral blood karyotype was performed using high-resolution G-bands by trypsin using Giemsa (GTG-banding), as described [11]. Ultrasonography and cystourethroscopy were performed by a paediatric radiologist and a paediatric urologist, respectively.

### 2.2. Next Generation Sequencing (NGS) and Filtering

The genomic deoxyribonucleic acid (DNA) was extracted from peripheral venous blood cells using the Gentra Puregene Blood Kit (Qiagen, Hilden, Germany). The DNA was quantified using a high-performance microvolume spectrophotometer Nanophotometer^®^ NP60 (Implen Inc., Westlake Village, CA, USA). Whole exome sequencing (WES) was performed by 3Billion, Inc. (Seoul, Republic of Korea). All exon regions of all human genes (~22,000) were captured by xGen Exome Research Panel v2 (Integrated DNA Technologies, Coralville, IA, USA). The captured regions of the genome were sequenced with Novaseq 6000 (Illumina, San Diego, CA, USA). We followed the best practice recommendations from the Broad Institute using the Genome Analysis Toolkit (GATK) for preprocessing, variant calling, and refinement. Raw sequence data were mapped to the 1000-Genomes phase II reference genome (GRCh37 version hs37d5) using the BWA-MEM algorithm of Burrows-Wheeler Aligner software, version 0.7.15-r1140. Duplicates were removed using Picard (Broad Institute). The variant call format file (VCF) was annotated using ANNOVAR [12]. Variant filtering and prioritisation were performed using B_platform (https://www.bitgenia.com/b-platform/, accessed on 24 June 2022). Candidate variants were selected when minor allele frequency (MAF) was <3% in gnomAD exomes and genomes and in 1000 Genomes. For further analysis, single nucleotide variants (SNVs) and indels with a read depth ≥ 10× and Genotype Quality (GQ) score ≥ 45 and variants with high and moderate impact on protein were filtered. The VarElect application (https://varelect.genecards.org/, accessed on 24 June 2022) was used to prioritise the variants based on the patient’s phenotype. Integrative Genomics Viewer (IGV v.1.4.2) [13] was used to visually inspect the variants. Human Genome Variation Society (HGVS) nomenclature was checked with Mutalyzer 3 [14]. We classified the variants according to their potential pathogenicity using the American College of Medical Genetics and Genomics/Association for Molecular Pathology (ACMG/AMP) guidelines for variant interpretation [15] and following the ClinGen Sequence Variant Interpretation Working Group (SVI WG) recommendations (https://www.clinicalgenome.org/working-groups/sequence-variant-interpretation, accessed on 24 June 2022). Additionally, applying the CNV prediction tool from NGS-derived data, DECoN (Detection of Exon Copy Number variants), we screened for potential CNV-type variants in phenotype-related genes [16]. The likelihood of nonsense-mediated messenger ribonucleic acid (mRNA) decay (NMD) was predicted using the NMDEscPredictor (https://nmdprediction.shinyapps.io/nmdescpredictor/, accessed on 24 June 2022).

### 2.3. Sanger Sequencing

The relevant variant identified in the proband was confirmed by Sanger sequencing of genomic DNA from the proband and his parents. *MYRF* exon 6 was amplified by polymerase chain reaction (PCR) with the specific primers (forward 5′-GCTTCCTGAAGGAGGTGTCC-3′, reverse 5′-AGCCTGTTTGCTCTTCTGTGA-3′) and GoTaq® DNA Polymerase (Promega, Madison, WI, USA). Products were sequenced using an ABI 3500 Genetic Analyzer (Applied Biosystems, Waltham, MA, USA) at the Translational Medicine Unit of the Buenos Aires Children’s Hospital (Unidad de Medicina Traslacional, Hospital de Niños Ricardo Gutiérrez, Buenos Aires). The sequences were compared to the reference sequence and analysed using BioEdit (BioEdit Sequence alignment editor) and Chromas (Chromas | Technelysium Pty Ltd., South Brisbane, QLD, Australia) tools. The following reference sequences were used: GRCh37 (human genome), MYRF: NG_047038.1 (gene), NM_001127392.3 (mRNA), NP_001120864.1 (protein).

## 3. Results

### 3.1. Clinical Observations and Diagnostic Testing

The proband was a full-term baby born in Argentina and referred to our Hospital for ambiguous genitalia at 6 days of life. Parents were non-consanguineous, and there was no remarkable family history. At the first visit, weight (3.580 kg), length (52 cm), and head circumference (34.2 cm) were within the normal ranges. The physical examination showed a phallus of 2 cm (length, −3.2 SDS) × 1.2 cm (width, 0.1 SDS), penoscrotal hypospadias, partially fused labioscrotal folds, an anogenital distance of 1.2 cm, and non-palpable gonads (EGS: 4/12). Abdomino-pelvic ultrasound, performed at 7 days of life, revealed a hypoechogenic tubular image (22 × 6 × 9 mm) with an echogenic central line, suggestive of a rudimentary uterus, and no gonadal structures were observed. A cystourethroscopy visualised a severe hypospadiac urethral meatus and a cavity compatible with a vagina. The bladder and bladder neck presented normal characteristics. The karyotype was 46,XY[30]. Hormonal laboratory testing, performed to assess the gonadal axis, showed that serum testosterone (16 ng/dL) and AMH (68 pmol/L) were low, whereas LH (20.2 mIU/mL) and FSH (6.88 mIU/mL) were high for age and chromosomal sex. These findings lead to the presumptive diagnosis of dysgenetic 46,XY DSDs.

### 3.2. Variant Filtering and Prioritisation

Filtering for candidate variants with MAF < 3% in gnomAD and 1000 Genomes and SNVs and indels with a read depth ≥ 10× and GQ score ≥ 45 among total exome variants yielded 1501 variants with high and moderate impact in 1133 genes. Subsequently, we selected candidate variants in genes potentially associated with the patient’s phenotype using the VarElect tool and prioritised 1 variant at chromosome 11 position 61539196, corresponding to exon 6 of *MYRF* (Figure 1). The variant was *MYRF*: NC_000011.9(NM_001127392.3):c.965G>A, NP_001120864.1:p.(Trp322*), indicating a stop-gain codon at position 322 of the protein and predicting an NMD mechanism. The position was read with a depth of 157×, with 81 reads for adenine and 76 reads for the reference allele, compatible with a heterozygous presentation. Sanger sequencing confirmed the existence of a de novo heterozygous variant in the proband, which was absent in both parents (Figure 2). The variant was not reported either in the consulted population databases (gnomAD exomes and genomes and 1000 Genomes) or in the literature. This variant has been reported to ClinVar (VCV001687379.1). The nonsense variant was classified as pathogenic according to the ACMG/AMP and ClinGen SVI WG recommendations with a score of 12 points: 8 points for PVS1 very strong (mRNA predicted to undergo NMD, and exon 6 of 27 is present in a biologically relevant *MYRF* transcript), 2 points for PM6 (Sanger sequencing confirmed that the variant was de novo but paternity analysis was not performed), 1 point for PM2 supporting (variant absent from all exomes and genomes of control individuals in the gnomAD exomes and genomes and 1000 Genomes databases), and 1 point for PP4 (the phenotype was specific to the condition, and the disorder has a limited number of genetic aetiologies with all those genes having been tested in the WES analysis for this patient).

Other variants in genes associated with the phenotype were ruled out due to insufficient evidence to support their pathogenicity (Appendix A). Copy number changes in sequences localised in known genome regions, which could be studied using array comparative genomic hybridisation (aCGH) or single nucleotide polymorphism array (SNP array), were ruled out directly using the NGS data: no clinically relevant deletions or duplications were prioritised through the DECoN-CNVs prediction algorithm (Appendix A).

### 3.3. Reverse Phenotyping

Once the *MYRF* variant was prioritised, a reverse phenotyping approach was used to rule out the involvement of other organs affected in patients described with reversible encephalopathy or OCUGS syndrome. An assessment of cardiac, pulmonary, ophthalmologic, intestinal, and central nervous systems was performed. In addition to the urogenital phenotype already described, which was the main complaint for our patient, we identified a meso/dextrocardia with situs solitus by ultrasonography. There was no structural cardiopathy, and the left ventricle systolic function was normal. No other phenotype was detected in the examined organs.

## 4. Discussion

In this work, we report the genetic diagnosis of a de novo, nonsense variant in the Pro domain of MYRF in a newborn with 46,XY DSDs, which drove to the detection of meso/dextrocardia through reverse phenotyping. Other malformations associated with *MYRF* variants were excluded. *MYRF* encodes myelin regulatory factor, a precursor of a transcription factor of the membrane-bound transcription factor (MBTF) type [17]. After the formation of a homotrimer, there is an autocatalytic cleavage in the Intramolecular Chaperone Auto-processing (ICA) domain that releases the N-terminal moiety of MYRF containing a proline-rich region (Pro domain) and a DNA-binding domain [18,19]. The N-terminal region of MYRF specifically activates the expression of myelin genes such as *MBP*, *MOG*, *MAG*, *DUSP15,* and *PLP1* during oligodendrocyte maturation [19]. Although MYRF is expressed in various other tissues, such as those derived from the coelomic epithelium, the target genes in those tissues have not yet been clearly elucidated [20]. 

The pathogenesis of DSD in the 46,XY patient may be explained by the fact that MYRF is normally expressed in the coelomic epithelium-derived cells [20]. Somatic cells of the testes responsible for androgen and AMH secretion are derived from the coelomic epithelium; when an early gene such as *MYRF* is not expressed, the differentiation of the testis is expected to be impaired resulting in gonadal dysgenesis and subsequent undervirilisation of the genitalia and persistence of Müllerian ducts. In 46,XX fetuses, blunted MYRF expression results in ovarian dysgenesis and disrupted differentiation of other derivatives of the coelomic epithelium, such as an absent uterus and Fallopian tubes [21].

With the development of NGS technologies, the application of the “phenotype-first” approach has led to the identification of variants in several new genes. Indeed, from clinical findings, the researchers have looked for variants in the genomes of the patients that may explain the symptoms. *MYRF* emerged as a candidate gene from the study of different cohorts of patients with urogenital, cardiac, and pulmonary phenotypes [22,23], congenital diaphragmatic hernia [21], reversible refractory epilepsy [24], or ophthalmopathy [25]. With the progressively decreasing costs of genomic studies, the “genotype-first” approach has gained ground, i.e., the detection of a genetic variant in a patient referred for a specific condition is followed by “reverse phenotyping” searching for clinically inapparent features associated with the genotype [26]. The finding of a nonsense variant in *MYRF* in the proband described here prompted the screening of other potentially affected organs in this rare aetiology for DSDs. No congenital anomalies such as those described in cardiac-urogenital syndrome (OMIM 618280) and encephalitis/encephalopathy, mild, with reversible myelin vacuolisation (OMIM 618113) were found in the lungs, diaphragm, central nervous system, or eyes. A mild congenital heart malrotation was observed: a thorough assessment indicated that the meso/dextrocardia was not associated with structural anatomic malformations, and cardiac function was normal. Therefore, a more severe cardiac phenotype, such as that described in MYRF-related ophthalmic cardiac urogenital syndrome (MYRF-OCUGS), could be ruled out [27].In summary, a good prognosis could be established for the cardiac phenotype, as opposed to what could have been predicted based on genetic testing eventually performed prenatally, given the existence of atypical genitalia.

Several reports in recent years provide evidence for the pleiotropic effect of MYRF, that is, a gene with multiple phenotypic expression. The variant type and localisation usually correlate with the resulting phenotype: the most deleterious (nonsense, frameshift, and splicing) variants spread along all MYRF domains have been associated with severe cardiac [22,23,28], lung [22,28], diaphragmatic [21], and urogenital [3,28] defects with early clinical manifestations. Missense variants in any part of the gene show milder phenotypes with later and/or transient manifestations [24]. Finally, variants in the C-terminal domain are associated with nanophthalmos with hyperopia [17,25]. In our patient, the variant localisation in the N-terminal domain of MYRF predicted the lack of ophthalmopathy. Conversely, due to the nonsense variant type predicting an NMD loss of function, a more severe phenotype could have been expected in the cardiac and respiratory systems. Our observation indicates that the spectrum of clinical manifestations cannot be predicted by the sole genotyping in patients with MYRF variants, and a thorough clinical assessment is warranted.

In conclusion, MYRF-associated phenotypes are complex, and patients that are initially studied by one specialty due to one or few cardinal signs/symptoms may subsequently require the attention of other specialties as the genetic aetiology is ascertained, leading to reverse phenotyping. Furthermore, multidisciplinary periodical follow-up may be necessary, given that hyperopia and encephalopathy could appear as late-onset manifestations. This represents an interesting model whereby genomic medicine drives clinical screening.

## Figures and Tables

**Figure 1 jpm-13-01158-f001:**
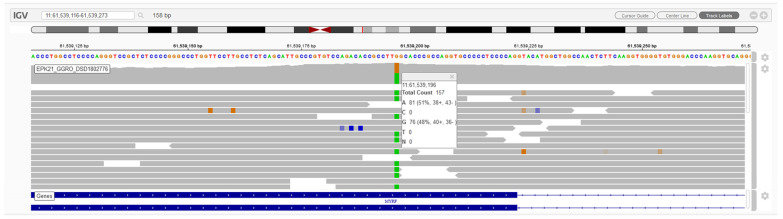
MYRF candidate variant by NGS. Integrative Genome Viewer (IGV) visualisation of variant MYRF: NC_000011.9(NM_001127392.3):c.965G>A. Green indicates reads of an adenine (A), brown of guanine (G), blue of cytosine (C) and red of thymine (T).

**Figure 2 jpm-13-01158-f002:**
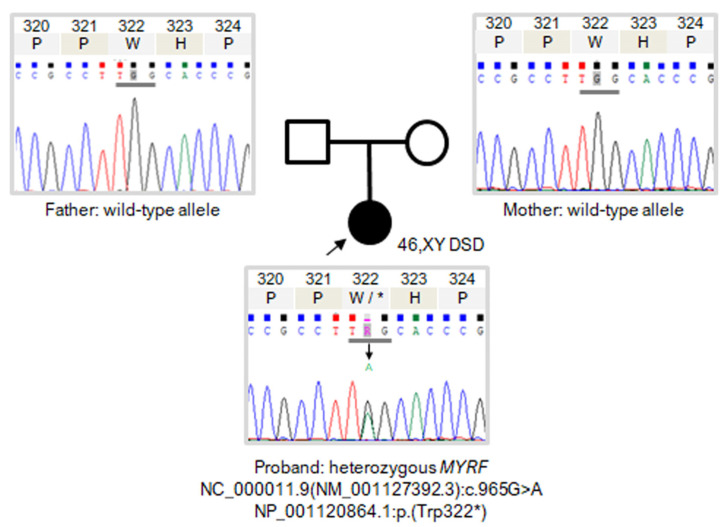
Family segregation results by Sanger sequencing: chromatograms of proband and parents. Pedigree: the arrow indicates index case; circle indicates phenotypic female; square indicates phenotypic male; and full black circle indicates complete phenotype (46,XY DSDs).

## Data Availability

Data on DNA sequencing are not publicly available due to privacy or ethical restrictions.

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
