# Peer review of "Identification of a Novel Variant in Myelin Regulatory Growth Factor by Next-Generation Sequencing Led to the Detection of a Clinically Inapparent Congenital Heart Defect in a Patient with a 46,XY Disorder of Sex Development"

_jpm, 2023, doi:10.3390/jpm13071158_

Round 1

Reviewer 1 Report

The authors report on full-term newborn presented with ambiguous genitalia and with 46,XY DSD in whom the WES identified de novo, heterozygous, nonsense variant in MYRF allowing to identify a clinically inapparent congenital heart defect by reverse phenotyping.

I must say that I really enjoyed reviewing this paper. Some small typing-corrections and/or minor revisions are necessary and I have some suggestions for further improvement of manuscript (these suggestions are not mandatory).

Minor comments

·       All abbreviations used in text should be reported in extended form when used for the first time (the same is when genes abbreviations are reported) besaide abreviation the name of the gene should be written in extended form with OMIM number.

should be written as : MYRF (Myelin Regulatory Factor; OMIM 608329)

·       Gene-Phenotype Relationships of MYRF gene should be reported: Cardiac-urogenital syndrome (OMIM 618280) and Encephalitis/encephalopathy, mild, with reversible myelin vacuolization (OMIM 618113).

·       All abbreviations used in text should be reported in extended form when used for the first time (the same is when genes abbreviations are reported) besaide abreviation the name of the gene should be written in extended form with OMIM number.

should be written as : MYRF (Myelin Regulatory Factor; OMIM 608329)

SuggestionS

·       It will be of interest to perform future multidisciplinary periodical follow up (pls. introduce this in the text and/or conclusion) in consideration of phenotype which could be changed and late onset manifestations  can be observed/valuated later in time pls. reported this in manuscript.

·       As I understund from reading, you have used only prediction tools for Variant interpretation.  It could be interesting to performe some functional genetic variation studies, aim to understand the molecular mechanisms and pathways that link genotype to phenotype or in silico modelling/analysis sounds like an interesting tool to consider in clinical diagnostic pipelines. If authors have the possibility to perform in silico analysis and presentation could be of interest. However if this is too complicate at least reported existance of this possibilities - state in manuscript .

·       Has any other Genetic Test been performed before WES (i.e. SNParray) in consideration of differential diagnosis? pls. explain/report in manuscript

Reviewer 2 Report

This is a well-written case report highlighting a relatively rare diagnosis.  I have the following comments/suggestions:

1. In the introduction, the authors note that in 46,XY individuals, insufficient virilization reflects either impairment in androgen secretion or action.  I would suggest changing the word “secretion” to “biosynthesis”.

2. In the Results section, is it necessary to note the city of birth?  This could narrow down the possibilities of who this patient is, given the history of ambiguous genitalia, which is not that common.

3. I would suggest changing the growth parameters at first visit from “height” to “length”, given that the Methods section suggests that the measurement was taken through an infantometer.

4. The authors note that they used a “genotype first” approach to diagnosis.  This is partially true, but their patient did have a phenotype which they used to lead them to the gene responsible for their patient’s features.  As an example, many DSD specific targeted multigene panels have MYRF on them, so a more targeted 46,XY DSD panel probably would have detected this variant too, as opposed to nontargeted whole genome sequencing. As such, they actually did use a “phenotype first” approach, albeit an incomplete phenotype, using NGS to prioritize genes that could have this partial phenotype (46,XY DSD) as a component.  They then completed the phenotyping based on the genetic diagnosis.  In the end, their patient had a clinically insignificant finding in the cardiovascular system only, so this didn’t really change the overall medical management of this patient.  As such, the authors may wish to refocus this report on the fact that their patient would have been predicted to have a severe phenotype but didn't, instead of focusing on completing the phenotyping as the main point of the case study.  Completing the phenotype is important and, in this case, led to the providers finding that he had a very good prognosis instead of a poor prognosis, which is what may have been predicted based on genetic testing results alone if this were to have been found in a prenatal setting.  Although the authors point this out at the end of the Discussion, this seems like the mail lesson here and is not otherwise highlighted in the case report.
